

# Potential limitations of using a modal aerosol approach for sulfate geoengineering applications in climate models

Daniele Visioni[1], Simone Tilmes[2], Charles Bardeen[2], Michael Mills[2], Douglas G. MacMartin[1], Ben Kravitz[3,4], and Jadwiga Richter[5]

[1]Sibley School of Mechanical and Aerospace Engineering, Cornell University, Ithaca, NY, USA
[2]Atmospheric Chemistry, Observations, and Modeling Laboratory, National Center for Atmospheric Research, Boulder CO, USA
[3]Department of Earth and Atmospheric Sciences, Indiana University, Bloomington, IN, USA
[4]Atmospheric Sciences and Global Change Division, Pacific Northwest National Laboratory, Richland, WA, USA
[5]Climate and Global Dynamics Laboratory, National Center for Atmospheric Research, Boulder CO, USA

**Correspondence:** Daniele Visioni (daniele.visioni@cornell.edu)

**Abstract.** Simulating the complex aerosol microphysical processes in a comprehensive Earth System Model can be very computationally intensive and therefore many models utilize a modal approach, where aerosol size distributions are represented by observations-derived lognormal functions. This approach has been shown to yield satisfactory results in a large array of applications, but there may be cases where the simplification in this approach may produce some shortcomings. In this work we show specific conditions under which the current approximations used in modal approaches might yield some incorrect answers. Using results from the Community Earth System Model v1 (CESM1) Geoengineering Large Ensemble (GLENS) project, we analyze the effects in the troposphere of a continuous increasing load of sulfate aerosols in the stratosphere, with the aim of counteracting the surface warming produced by non-mitigated increasing greenhouse gases concentration between 2020-2100. We show that the simulated results pertaining to the evolution of sea salt and dust aerosols in the upper troposphere are not realistic due to internal mixing assumptions in the modal aerosol treatment, which in this case reduces the size, and thus the settling velocities, of those particles and ultimately changes their mixing ratio below the tropopause. The unnatural increase of these aerosol species affects, in turn, the simulation of upper tropospheric ice formation, resulting in an increase in ice clouds that is not due to any meaningful physical mechanisms. While we show that this does not significantly affect the overall results of the simulations, we point to some areas where results should be interpreted with care in modeling simulations using similar approximations: in particular, the evolution of upper tropospheric clouds when large amount of sulfate is present in the stratosphere, as after a large explosive volcanic eruption or in similar stratospheric aerosol injection cases. Finally, we suggest that this could be avoided if sulfate aerosols in the coarse mode, the predominant species in these situation, are treated separately from other aerosol species in the model.

## 1 Introduction

A comprehensive representation of aerosol processes in Earth System models is crucial for a variety of reasons. Aerosols are one of the main short-term forcing agents in the climate system, and uncertainties in the estimate of their overall forcing effects



are still quite large Boucher et al. (2013). Directly, they scatter incoming solar radiation, thus influencing surface temperatures, and also absorb both solar radiation and outgoing planetary radiation, locally increasing air temperatures, modifying circulation patterns, and affecting meteorology. Indirectly, they affect the climate by modifying cloud cover, acting as cloud condensa-

tion nuclei (CCN) and ice nuclei (IN), and changing clouds' physical and radiative properties (also through local atmospheric heating). Lastly, once they are deposited to the surface, they affect land and ice albedo through the melting of snow and ice. Aerosols at the surface also contribute to air pollution (in particular with particulate matter (PM) below 2.5 $\mu$m and 10 $\mu$m in diameter, Ayala et al. (2012)) and may affect soils and ecosystems through acid deposition (Vet et al. (2014)).

Atmospheric aerosols may have different sizes, ranging from 0.001 $\mu$m to 100 $\mu$m in diameter, and their characteristics (number concentration, mass, shape, chemical composition and other physical properties) may change through emission (from natural or anthropogenic sources), nucleation (defined as the formation of new particles), coagulation (defined as the combination of existing aerosol particles, decreasing their number concentration but leaving the overall mass unaltered), condensational growth of chemical species in vapor form (such as $H_2SO_4$, $NH_3$, $HNO_3$ and volatile organics gases) on existing particles, gas-

phase and aqueous-phase chemistry, water uptake (Ghan and Zaveri (2007)) and their removal through gravitational settling (dry deposition), in-cloud scavenging (defined as the removal of aerosol particles by precipitation particles), below-cloud scavenging (defined as the capture of aerosol particles by precipitating droplets, Feng (2009)). They can, of course, be composed of different chemical species: the main components are usually sea salt, mineral dust, black carbon, organic matter, nitrate, ammonium and sulfate. Wang et al. (2020) provide a recent overview of all aerosol-related processes: this list of processes can

give an idea of how challenging it can be to represent aerosol processes correctly in climate models, both due to the uncertainties in each process and in their combination, and because of the high computational burden necessary to correctly reproduce the known parts of each process. For this reason, most processes need to treat aerosols in parametrized ways with different complexities, ranging from the least (bulk model, considering all aerosols as described by a single mean radius and standard deviation) to the most complex (sectional model, considering a large number of "bins" of different sizes for each species where

the aerosols can grow or shrink). An intermediate approach is the modal method: it works on the assumption that the actual size distribution of the aerosols can be represented by the combination of multiple log-normal functions with fixed standard deviations that are based on observations. Furthermore, a modal approach to aerosol microphysics must also make assumptions about the internal and external mixing of aerosol species (Fassi-Fihri et al. (1997)): internal mixing is defined as the modeling of an aerosol particle as a mixing of the constituent species (Clarke et al. (2004)), either as an homogeneous mix or as a coated

sphere containing a solid core and coated by a liquid exterior. More simply put, it assumes that different species of aerosols in every gridbox are present in the same proportion, and that their physical characteristics can be described by the same average size distribution. External mixing is defined as the treatment of aerosol particles as composed of different species, and no assumption needs to be made with regards to the localized proportion of the various species, as they are treated differently.

A modal treatment of aerosols in climate models has been shown to successfully reproduce aerosol measurements in various occasions. In this work, in particular, we focus on the Community Atmosphere Model version 5.0 (CAM5) and its implemen-





tation in the Community Earth System Model (CESM), using the Modal Aerosol Module with 3 modes (MAM3, Liu et al. (2012)). This approach has been shown to correctly reproduce tropospheric aerosols in a baseline climate (see Liu et al. (2012); Samset et al. (2014)), and also to correctly reproduce the evolution of the stratospheric sulfate aerosol layer in the extreme case

of explosive volcanic eruptions (Robock (2000)), as detailed by Mills et al. (2016, 2017) for the Whole Atmosphere Community Climate Model (WACCM) version of CESM1, with a high top (140 km) and 70 vertical layers. This model has been used for climate simulations of Stratospheric Aerosol Intervention (SAI), a form of climate engineering that has been proposed (Crutzen (2006)) as a way to temporarily reduce global surface temperatures by mimicking the cooling effect of volcanic eruptions through injecting $SO_2$ in the stratosphere. In both the case of volcanic eruptions and for proposed artificial injections,

once $SO_2$ reaches the stratosphere, it would oxidize, eventually forming gaseous $H_2SO_4$, which would then either nucleate forming new sulfate aerosol particles of $H_2SO_4$-$H_2O$ or condense onto existing particles (already existing particles could also coagulate, resulting in larger particles with the same overall mass). While this process transforms all $SO_2$ into aerosols with an e-folding time of around 1 month (Mills et al. (2017)), the produced aerosols tend to remain in the stratosphere for one year or more (Visioni et al. (2018b)), and are removed through large-scale stratospheric circulation, that moves them poleward, or by

gravitational settling or stratosphere-troposphere exchange, crossing the tropopause. Once in the troposphere, they are quickly removed through dry or wet deposition (Kremser et al. (2016)).

Climate engineering simulations with CESM1(WACCM) in the Geoengineering Large Ensemble (GLENS; Tilmes et al. (2018a)) have shown that it would be possible to maintain global surface temperatures at 2010-2029 levels even under a scenario where emissions of greenhouse gases continue unabated, by increasing the amount of $SO_2$ injected throughout the

century. This technique may be able to reduce some of the harmful climatic effects produced by the temperature increase ((Tilmes et al. (2018a); Kravitz et al. (2019))), by reducing the amount of incoming sunlight, but would not be a perfect solution. In fact, the atmosphere and surface would be impacted in various ways by the produced aerosols, for example, the chemical composition of the stratosphere and the dynamical response produced by the local stratospheric heating (Richter et al. (2017); Tilmes et al. (2018b)), which in turn can influence precipitation (Simpson et al. (2019)) and the high-latitude

seasonal cycle of temperature (Jiang et al. (2019)). Once the aerosols are deposited to the surface they may also affect soils; however, considering the much larger amount of tropospheric sulfate aerosols produced by both natural and human activities, those settling down from the stratosphere would have a marginal impact everywhere except in some pristine areas (Visioni et al. (2020b)). Other processes that may be affected by SAI include aerosol interactions with cirrus clouds, which are a key component of the radiation balance. Water clouds at lower altitudes have a net cooling effect because they reflect solar radiation

(see for instance Yi et al. (2017)); on the other hand, the effect of cirrus clouds made of ice crystals and produced in the upper troposphere by supercooled water particles is harder to determine, but is widely understood to be positive (i.e., it produces a net warming at the surface) (Fusina et al. (2007)) due to their trapping of outgoing planetary radiation. Kuebbeler et al. (2012) and Visioni et al. (2018a) found in two different climate models (ECHAM-HAM5 and ULAQ-CCM, using a modal and a sectional aerosol approach, respectively) that the change in vertical temperature gradient resulting from the stratospheric heating would

reduce the formation of tropical cirrus ice clouds by less than 10%, thus contributing to the surface cooling. This effect is tied to the fact that, in both models, the amount of water vapor reaching the upper troposphere and necessary for the cloud





formation is directly tied to the available turbulent kinetic energy, which is a function of the vertical temperature gradient, and is therefore a purely dynamical effect. Cirisan et al. (2013) on the other hand investigated the possible microphysical effect of the increased sulfate load on the cirrus cloud formation, but found no significant impact due to the much larger size of the
aerosols from the stratosphere compared to those already present in the troposphere by other activities.

In this study, as an example to illustrate some of the shortcomings of the modal aerosol treatment in MAM3, we use the GLENS simulations to explore the effects of large amount of sulfate aerosols in the stratosphere on tropospheric aerosol concentrations and on cirrus ice cloud formation in CESM1(WACCM). In the following sections, we will briefly describe how aerosol microphysics and cirrus ice formation are parameterized in Section 2, then discuss how aerosols in the upper
troposphere change in the simulations of interest in Section 3 and how that affects upper tropospheric ice in Section 4. Finally, possible radiative effects at the top of the atmosphere of the observed changes will be discussed in Section 5.

## 2   Model description

In this section we will briefly describe the simulations used in this work, and then describe the components of the model that will be of use in our analyses.


The Geoengineering Large Ensemble (GLENS, Tilmes et al. (2018a)) is an ensemble of simulations performed with CESM1 (WACCM), with all simulations using surface emissions from the Representative Concentration Pathway 8.5 (RCP) scenario. 21 ensemble members are available for the period 2010-2030 under RCP8.5. From each of these, in 2020 a scenario is simulated where $SO_2$ is injected at four locations: 30° N and S, with injections at 23 km altitude, and 15° N and S at 25 km altitude.
Each year, an algorithm (described by Kravitz et al. (2017)) determines the amount of $SO_2$ to be injected at each location in order to maintain the mean surface temperature, the inter-hemispheric surface temperature gradient, and the equator-to-pole temperature gradient at their 2020 values in the presence of growing greenhouse gas concentrations. All the simulations of SAI are extended to 2100 (hereafter, termed GLENS), and 4 ensemble members of RCP8.5 without SAI are run to the year 2097 (hereafter, termed Baseline).

### 2.1   The Modal Aerosol Model in CESM

The Modal Aerosol Model (MAM) has been first described by Liu et al. (2012), and therein evaluated for tropospheric aerosol loads. Some modifications have been made for including interactive stratospheric aerosols, described in depth by Mills et al. (2016) and therein validated in the case of the Pinatubo 1991 eruption.

For CESM1(WACCM) climate simulations, the 3-mode version (MAM3) is used, with aerosol species divided in three dif-
ferent lognormal modes of fixed width, named Aitken (dry diameter size range between 0.015 and 0.05 $\mu$m), Accumulation (between 0.05 and 0.3 $\mu$m) and Coarse mode (between 0.80 and 3.65 $\mu$m; all estimates from Liu et al. (2012)), going from the smallest to the largest (in this manuscript, we will always refer to the names of the modes with a capitalized letter to avoid ambiguity). Sulfate particles can grow by condensation (from $H_2SO_4$ vapor condensing on existing particles, locally maintain-





ing particle numbers but increasing mass) or coagulation (locally reducing particle numbers but maintaining mass). When, by
either process, the tail of the distribution of the particles in one of the modes grows to a size that would nominally be in the
size range for the larger mode, the particles are transferred to the larger mode. This is done, as detailed by Easter et al. (2004),
by defining a lower and upper limit for the dry diameter in each mode, and transferring part of the local number concentration
to the larger mode when the threshold is surpassed. In the stratosphere the sulfate particles can also shrink due to evaporation,
thus allowing for a Coarse-to-Accumulation mode transfer.


Compared to the more computationally-demanding MAM version with 7 aerosol modes (MAM7), in MAM3 for purposes
of size and number concentration calculations only one type of particles for each mode is considered, rather than having them
differentiated by aerosol species (i.e. different chemical composition). The mass for the single species has to be conserved,
both globally and locally, and thus can only change in each gridbox if particles are moved from one gridbox to the other, either
because of air mass movement, or because of gravitational settling or other tropospheric removal processes. Therefore, each
particle species shares the same radius and number concentration per each mode, but the overall mass of the single species can
vary. Liu et al. (2012) justify this approach by noting that the sources of different aerosol types are geographically separated,
thus unlikely to affect each other in the simplified version used for long climate applications. Coarse mode sulfate aerosols
are, in quiescent conditions, also scarce in the troposphere: in the AeroCom multi-model mean (Textor et al. (2006)) they
were determined to be less than 2% of all sulfate aerosols, with a preponderance of particles in the Accumulation mode in the
troposphere. The assumption by Liu et al. (2012) would therefore hold in the background atmosphere. We will show in the
next section that the presence of the Coarse mode sulfate particles produced by SAI fundamentally breaks this assumption,
unnaturally modifying the size and quantity of non-sulfate aerosol species in the upper troposphere.

### 2.2 The formation of cirrus ice clouds in CESM

Cirrus clouds play an important part in the radiation budget, but have been generally represented poorly in general circulation
models (GCMs) for a variety of reasons, amongst them a poor horizontal resolution which fails to capture the scale needed to
represent some of the processes and a large spread in the ice water content simulated by models (Jiang et al. (2012)). There
are two processes that can produce ice crystals in the upper troposphere: homogeneous nucleation of ice crystals from sul-
fate aerosols and heterogeneous immersion freezing of mineral dust. Normally, homogeneous freezing is assumed to be the
dominant process, although there are instances where this is not the case (Knopf and Koop (2006); Cziczo et al. (2013)). In
the atmospheric model used in CESM1(WACCM), the Community Atmospheric Model version 5 (CAM5), both processes are
present and we will briefly describe both below.

The process of homogeneous freezing is based on the assumption that only sulfate particles in the Aitken mode work as ice
nuclei (IN), using the portion of the Aitken mode particles with radii greater than 0.1 microns (Liu and Penner (2005); Liu et al.
(2007)). Other works have used all available sulfate modes for homogeneous freezing (Shi et al. (2015)), and this would clearly
affect the results. The process of homogeneous freezing is assumed to happen only when clouds are present, together with a





probability distribution from Kärcher and Burkhardt (2008) that determines when the supersaturation is above the threshold for homogeneous freezing to happen. This means that homogeneous freezing can happen only when local vertical velocities are high (Kärcher and Lohmann (2002)). Those velocities are determined following Morrison and Pinto (2005) as

$$w_{sub} = \sqrt{\frac{2}{3}TKE} \tag{1}$$

with the turbulent kinetic energy (TKE) determined using a steady state energy balance as in Bretherton and Park (2009).

For heterogeneous freezing, only Coarse mode dust particles are assumed to be available IN (although the exclusion of soot particles and other particles as inefficient ice nuclei has been debated; see Kärcher and Burkhardt (2008)). Given the internal mixing assumption in MAM3, the number of available dust nuclei is determined as a fraction of the overall amount of Coarse mode particles given the mass of dust and the overall aerosol mass

$$N_d = \left(\frac{m_d}{m_d + m_{ss}}\right) \times N_C \tag{2}$$

where $m_d$ is the mass of dust in the Coarse mode, $m_{ss}$ is the mass of sea salt in the Coarse mode, and $N_c$ is the total number of particles in the Coarse mode. This approach assumes a negligible amount of Coarse mode sulfate in the upper tropical stratosphere in the denominator of the fraction. Another shortcoming of the heterogeneous freezing description in this version of WACCM, as already discussed by Mills et al. (2017), is that when heterogeneous freezing occurs, IN that have nucleated to form ice particles are not removed from the available reservoir, thus allowing for too many particles to be formed via heterogeneous freezing.

This approach to the microphysical modeling of cirrus clouds in CESM has been discussed recently by Maloney et al. (2019), comparing it with a more complex approach using the coupled Community Aerosol and Radiation Model for Atmospheres (CARMA) sectional scheme, and comparing both with measurements from the National Aeronautics and Space Administration Airborne Tropical Tropopause Experiment (ATTREX 3) and with observations from the Cloud-Aerosol Lidar with Orthogonal Polarization (CALIOP) onboard the CALIPSO satellite. They found that while the CAM5 approach was capable of correctly representing the annual average cloud fraction profile in the tropics (20°N to 20°S), it tended to underrepresent the cirrus fraction in the tropical tropopause layer. Similar results where found previously also by Bardeen et al. (2013).

## 3 Simulated tropospheric aerosols

The geographical distribution of tropospheric aerosols in the unperturbed atmosphere depends mainly on the surface sources of the aerosols or aerosol precursors; once the aerosols are produced in the atmosphere or are directly emitted, they can be affected by long range transport, upward currents and sediment through gravitational settling or scavenging. An in-depth analyses of sources and sinks of atmospheric aerosols can be found, for instance, in Lamarque et al. (2010). We focus here on sulfate, dust and sea salt, as their changes will be of interest when analyzing the effects on cirrus formation. Sulfate particles are mostly formed through surface emission of $SO_2$, which can then oxidize and form sulfate particles of sulfuric acid in the smaller (Aitken) mode. For this reason, Coarse mode sulfate particles near the surface are much less frequent than dust or sea salt,





as shown in Fig. 1a. For the latter two, their concentration decreases with increasing altitude as the scavenging processes

reduce their number, whereas there is an additional sulfate layer in the stratosphere (the Junge layer), discovered and discussed

by Junge et al. (1961). This happens because of the presence of various sources of sulfate aerosols in the stratosphere, even

without considering volcanic aerosols from explosive volcanic eruptions. In particular, surface emissions of carbonyl sulfide

(OCS) and dimethyl sulfide (DMS), which are light, well mixed gases that may reach the stratosphere where they are oxidized,

forming sulfate aerosols (Vet et al. (2014)). Meteoric sulfur also plays a part in the formation of the layer (see Gómez Martín

et al. (2017)). The quantity of sulfate aerosols produced in the stratosphere with an artificial injection of $SO_2$ would be larger

by some orders of magnitude compared to the amount in the quiescent Junge layer; the full stratospheric distribution in the

case of GLENS has been described elsewhere (i.e., Kravitz et al. (2019)), so here we simply report that at 50 hPa, between

30°N and 30°S, the average mass concentration for the last 20 years of simulation is 163 $\mu$g/kg, compared to 0.4 $\mu$g/kg in the

unperturbed case; while this simulation considered high cooling ($\sim 4$°C reduction in global mean temperature), even much

smaller amounts of SAI would lead to substantially more stratospheric sulfate than the unperturbed case. This magnitude is

driven by the long lifetime of the produced aerosol particles in the stratosphere (around 12 months, Visioni et al. (2018b)).

Once the particles cross the tropopause, the various removal processes strongly reduce the lifetime, driving the concentration

down. This is visible already in Fig. 1a, with the tropopause clearly visible as a change in concentration. Nonetheless, the

average concentration of sulfate under SAI is much larger even in the upper troposphere (Fig. 1b) and only returns close to the

Baseline levels close to the surface, where other sulfate sources are predominant (Fig. 1c).

Fig. 1 shows the behavior of dust and sea salt in the Coarse mode in the same altitude-latitude region (those are the only

other aerosol species considered in this version of CESM1 in the Coarse mode: black carbon is only found in the Accumulation

mode). Various behaviors can be observed depending on the altitude of analyses for the GLENS case. We focus here on the

uppermost troposphere, right below the tropopause (black boxes in Fig. 1) and on on a layer below, in the upper troposphere

(green boxes). In the upper level, both dust and sea salt present a behavior that is much more similar to that of sulfate: a large,

initial increase in concentration in the first 5 to 10 years followed by either a constant evolution or by a small decrease (from

0.1 to 0.2 $\mu$m/kg for dust in the upper layer, from 0.06 to 0.2 $\mu$m/kg and from 0.9 to 0.17 $\mu$m/kg for sea salt in the upper and

lower layer). Only dust in the lower layer does not show such an abrupt initial increase. On the other hand, in the Baseline case

the dust mixing ratios show a decrease by 25% over the 80 years of analyses in the lower level. There is no immediate physical

mechanism by which such changes would be observed in aerosol species independent of sulfate; therefore, a more in-depth

analysis is necessary.

One possible explanation would be a change in the surface emissions of these species. However, the vertical profiles of

tropical concentration for the three species seem to exclude that. This is further confirmed by the analyses of the overall bur-

dens shown in Fig. 2. For these, no initial abrupt change can be observed, and the time evolution shows the opposite behavior

from that observed in Fig. 1, with a minor reduction in GLENS compared to Baseline. The behavior of the Baseline case is

consistent with previous projections: Mahowald and Luo (2003) predicted a reduction in overall dust emission with increasing

GHGs due to higher precipitation and more surface moisture produced by the warmer air; thus the precipitation decrease in

GLENS (Cheng et al. (2019)) would exacerbate differences between dust burden in GLENS and the baseline. Struthers et al.





(2013) on the other hand projected a small increase in sea salt aerosols mostly as a result of increased surface wind speeds.

Having excluded that changes in surface emissions are the cause of our observed abrupt increase, we further investigate the smaller aerosol modes (Fig. 3). In this case, we see a much closer agreement of the values in GLENS to those of the initial baseline conditions for sea salt and dust: this indicates that the initial aim of the GLENS simulations, which was to maintain the state of the climate as close as possible to 2010-2029 conditions, also helps with not modifying surface emission sources of those species. The time evolution of the globally averaged quantities for these species and modes also doesn't show the

same abrupt change as that observed in Fig. 1 (see Fig. S1 and S2). So clearly the cause of the rapid change in concentration of the Coarse mode must be the result of the very rapid increase in $SO_4$ aerosol concentration by more than 4 orders of magnitude.

The solution to this conundrum can be found by analyzing the behavior of the simulated radius of the particles in the three modes (Fig. 4). In MAM3, all aerosol species are assumed to be internally mixed. The simplified assumption that different

aerosols peak at different locations in the lower atmosphere is reasonable for the baseline scenario (Fig. 1 for the vertical distribution, and Fig. 2 for the spatial distribution). The parameterization requires that in each gridbox, all aerosols in one mode are treated as the same in terms of their size distribution, so for each mode, only one mode diameter and one number concentration is used in the online calculations (since the geometric standard deviation is constant throughout the atmosphere). The mass concentrations of the different species are then calculated using a reference density for each species (Liu et al.

(2012)). This information is necessary to calculate the gravitational settling velocities (at each layer), which follow Seinfeld and Pandis (2016), where the equation of a free-falling spherical particle in a fluid which has reached terminal velocity (which is done in less than $1 \times 10^{-6}$ for particles of diameter of $\simeq 1$ $\mu$m) is shown to be:

$$v_t = (\frac{4gD_pC_c\rho_p}{3C_D\rho})^{1/2} \tag{3}$$

where $D_p$ is the diameter of the particle, $\rho_p$ is the density of the particle, $C_c$ is the slip-correction factor that accounts for

non-continuum effects (dependent on the diameter of the particles), and $C_D$ is the empirical drag coefficient (dependent on the Reynolds number $R_e$).

The initial observed changes can thus be explained this way. The $SO_4$ formed in the stratosphere is the predominant form of aerosols in the stratosphere, deciding the radius of the particles at those altitudes. Most of these particles are in the Coarse mode, therefore changing the radius considered by the model in those grid-boxes. Once the sulfate aerosols cross the tropopause

(whether by gravitational settling, mostly in the tropics, or by large scale circulation, at higher latitudes) they are added to the particle distribution already present in the upper troposphere. Due to the small amount of Coarse mode aerosols there, and due to the internal mixing assumption in MAM3, the dust and sea salt aerosols already present are "forced" to reduce in size (and to conserve mass, to increase in number concentration, see Fig. S3). This produces a drop in the overall size by around 25% within 3 years in the upper troposphere. Because of the strong dependence of gravitational settling velocities on size, the result

is a drop in these velocities that, for dust aerosols, can be estimated to be over 40%. This is not a direct output of the model, but we estimate it from Eq. 3 using the approach of Seinfeld and Pandis (2016). The clear drop in $v_t$ can then explain the initial observation of increase in non-sulfate aerosol species: reducing settling deposition abruptly directly affects the concentration



**Figure 1.** Mass concentration (in $\mu$g per kg of air) of Coarse mode species (Sulfate, Dust and Sea Salt, from top to bottom). On the left, the Baseline (2020-2029) zonal mean concentration, with the black box indicating the area considered in the average for the central panels, where the annual mean evolution for both cases is shown. Lighter lines show the single ensemble realizations, while the thicker lines show the ensemble mean. On the right, the vertical profile in both cases in the same latitudinal band is shown. Black dashed lines in the central panels indicate the periods of analyses for the left and right panels; the same lines in the right panels indicate the altitude of analyses in the central panels.

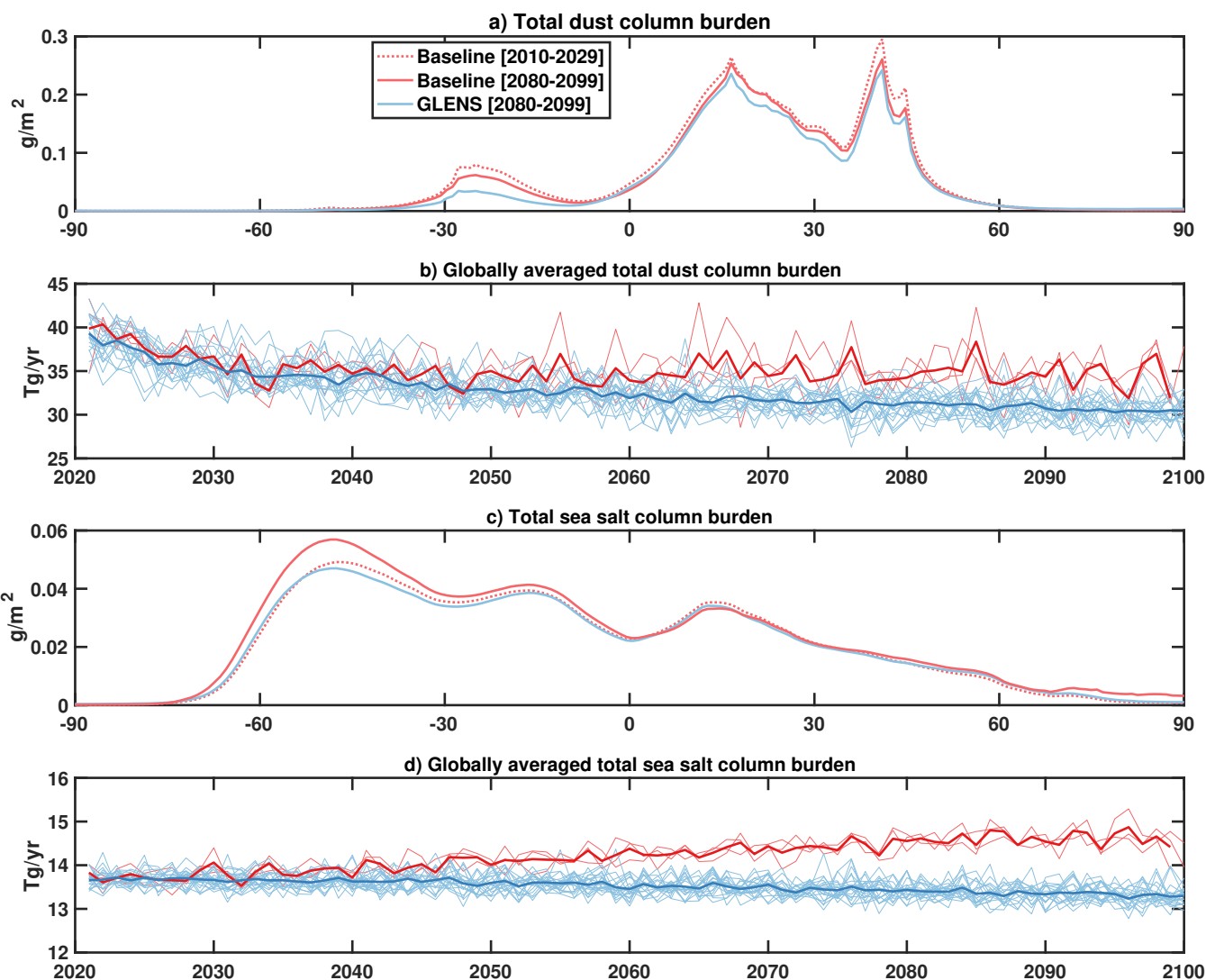

**Figure 2.** Zonally averaged total column burden of dust (a) and sea salt (c) (considering all modes) in 2010-2029 (red, dashed line), 2080-2099 in Baseline (red line) and 2080-2099 in GLENS (blue line). Evolution of the globally integrated dust (b) and sea salt (d) burden (Tg/yr). Lighter lines show the single ensemble realizations, while the thicker lines show the ensemble mean.



**Figure 3.** Mass concentration (in $\mu$g per kg of air) of Sulfate, Dust and Sea Salt in the Aitken (smallest) and Accumulation (intermediate) mode, between 30°N and 30°S. Lighter lines show the single ensemble realizations, while the thicker lines show the ensemble mean.



**Figure 4.** Wet radius (in $\mu$m) of all three modes in the Baseline case ([2010-2029], a-b-c-d) and in GLENS ([2080-2099], e-f-g-h). g) Vertical distribution of aerosol size in all three modes between 30°N and 30°S. Black boxes in panels c-f) indicate the area analyzed in the average for panel h), where the temporal evolution in Baseline (red) and GLENS (blue) is shown. Lighter lines show the single ensemble realizations, while the thicker lines show the ensemble mean. i) Estimate of the changes in gravitational settling velocities following Seinfeld and Pandis (2016).

by decreasing removal, especially in regions where contributions from below are scarce. And the reason why this is only visible in the upper troposphere is that, further below, the preponderance of pre-existing, background Coarse mode aerosol reduces

the strength of this effect, not changing the radii. The preponderance of other removal mechanisms (in-cloud scavenging) also reduces the effect of the gravitational settling changes.




## 4  Effect on cirrus clouds formation

We have shown that the simulated changes in dust and sea salt in the upper troposphere in GLENS using CESM1(WACCM), are non-physical in nature. However, by looking at the concentration of the same species in the Baseline conditions it is clear that the overall presence of those aerosols at the altitudes of analyses is small, and therefore would have a negligible effect both when looking at the direct effects (for instance, on the radiative fluxes, which would be overshadowed by the effects of the sulfate aerosols in the stratosphere) and when eventually looking at surface effects. We show in this section, however, that their change does influence the simulation of ice clouds formation in the upper troposphere. Visioni et al. (2021) showed that some noticeable changes in cloud cover are present in the GLENS simulations, and that when separating the contribution of different types of clouds, most of these changes were attributable to changes in high clouds, that are defined in CESM1(WACCM) as all clouds formed between the altitude of 400 and 50 hPa. Furthermore, these changes were not as noticeable in other performed simulations where the same cooling as GLENS was achieved using a solar constant reduction approach, and when together with that the stratospheric heating produced by the aerosols was imposed, without the presence of the aerosol themselves. This would point towards a contribution to the observed changes in high cloud cover in GLENS by some effects connected to the aerosol themselves. In particular, this points to some changes in the freezing processes that produce ice crystals in the upper troposphere, given that at those altitudes there is much less water than in the lower troposphere, and most of that is in the form of ice rather than liquid droplets (see Fig. S5).

As we detailed in Section 2.2, there are two types of processes that may lead to the formation of ice crystals at those altitudes: the spontaneous freezing of small aqueous sulfate droplets at high relative humidity rates (Chen et al. (2000)), known as homogeneous freezing, and the freezing of water droplets mediated by the presence of insoluble aerosol particles (ice nuclei, IN), known as immersion or heterogeneous freezing (Diehl and Wurzler (2004)), which can happen at lower relative humidity conditions and higher temperatures (Knopf and Koop (2006)). Due to the difficulties in measuring the amount of ice crystals in the upper troposphere, and to the challenges of representing the processes in models, there are plenty of uncertainties over the predominance of one of the two formation processes over the other. In CESM, only sulfate particles in the Aitken mode can act as the substrate over which homogeneous freezing can take place; this is generally understood to be correct, as too large particles would be increasingly harder to freeze (Chen et al. (2000)). Since Aitken mode particles do not change in GLENS (see Fig. 3a), changes in homogeneous freezing due to that can be excluded, as predicted already by Cirisan et al. (2013). However, changes in homogeneous freezing may happen as a result of dynamical changes in the local vertical velocities that determine the amount of water vapor in the upper troposphere. Since the processes that determine those local vertical velocities happen at a much smaller scale compared to the resolution of climate model, they are usually parameterized in climate models as a function of vertical stability (see Eq. 2.2). Both previous studies analyzing the response of ice clouds in a SAI scenario, Kuebbeler et al. (2012) and Visioni et al. (2018a), used climate models with a similar parameterization as CESM, and observed a reduction in ice cloud coverage because the warming of the stratosphere, combined with a cooling of the surface, produced a reduction in the vertical temperature gradients and thus in the turbulent kinetic energy used to determine the velocities. A similar process



can then be expected in CESM. Lastly, there would be no physical reason why SAI would change heterogeneous freezing in ice clouds. However, given what we observed in the previous section related to dust aerosols in the upper troposphere, an influence on freezing processes cannot be excluded.

In Fig. 5 we show an analyses of the changes in the amount of ice in clouds under GLENS. Two particularly different behaviors can be observed when separating the effects at low latitudes (30°N-30°S) versus the effects observed elsewhere. At low latitudes, the dynamical changes produced by the different vertical stabilities dominate, resulting in a slow decrease in ice concentration in line with predictions discussed by Kuebbeler et al. (2012) and Visioni et al. (2018a) (which is not a surprise, considering the very similar parameterization of sub-grid vertical velocities). At higher latitudes, where there are fewer changes in the vertical temperature gradient (since the stratosphere warms much less, see Richter et al. (2017)), the predominant effect is the sudden increase in IN, resulting from the simplified aerosol treatment in the Coarse mode. The increase in ice formation the model "sees" is therefore not physical in origin, but simply an artifact of the microphysical parameterization and thus should not be treated as a physical side-effect to SAI. A clue as to the two different mechanisms at play can be determined by observing the rate of change at low versus high latitudes. At low latitudes, the changes happen gradually as the stratospheric sulfate load (and thus the warming) increases. At high altitudes, the in-cloud ice changes are very abrupt, much more similar to the aerosol changes observed previously in this work. Further confirmation can be derived by observing the differences between the full GLENS simulations and the simulations described by Visioni et al. (2021) with stratospheric heating imposed, but no aerosols. By comparing the two (see bottom panels of Fig. 5), the effect of the presence of the aerosols can be separated from the dynamical ones, and indeed this confirms the different nature of the observed ice changes, since the two simulations (aerosols+stratospheric heating and stratospheric heating alone) show no differences at low latitudes, whereas they are as different as GLENS-Baseline at high latitudes.

Further information would be gained by observing the changes in heterogeneous versus homogeneous freezing in these simulations. However, output fields that separate the two processes have not been saved in the original simulations from GLENS. Therefore, a complete analysis is not possible using the whole ensemble and time period. Noting that the main, unexplained processes (with our current understanding of ice formation) happen in the very first decade of the simulations, for the following analyses we have re-run the first 21 years of one of the ensemble members with the same configuration (thus bit-by-bit, the result is the same) but retaining information related to the two different upper tropospheric freezing processes. The results in Fig. 6 confirm our previous observations. For homogeneous freezing, very few changes are initially observable at all latitudes, whereas large changes are observable in the concentration of what the model considers to be IN for heterogeneous freezing, sharply in the first few years. This increase is much larger than the increase in Coarse mode dust aerosols observed in Fig. 1, where the overall concentration was doubled, while here the increase is much larger, at more than 10 times the amount in the first two years. To partially explain this change we refer to the equation used by the model to determine the fraction of the available dust in the Coarse mode, which is used as IN for heterogeneous freezing (Eq. 2.2). The available number of dust particles that can be used by the model depends on the overall amount of Coarse mode particles, but is weighted by which fraction, locally, of aerosol is dust compared to sea salt. This formula (which in normal conditions gives satisfactorily results, see Maloney et al. (2019)) presumes a lack of sulfate in the upper troposphere. In our case however, as we have shown,



the amount of Coarse mode sulfate is actually by far preponderant in the upper troposphere. So while the number of Coarse mode particles $N_C$ grows (see Fig. S3), the fraction of all aerosols that is considered does not. This is a separated, but similar, problem as the one observed for the behavior of the aerosols. We tested how sensitive this assumption is by performing an identical simulation to the one that gave us information about homogeneous and heterogeneous freezing, but modifying the

code so that Eq. 2.2 becomes

$$N_d = \left(\frac{m_d}{m_d + m_{ss} + m_s}\right) \times N_C \tag{4}$$

thus correctly accounting for the mass of sulfate when considering which fraction of the number of Coarse mode particles locally present is made of dust aerosols. We show the results of these sensitivity simulation in Fig. 6 using dotted lines: as expected, this doesn't change the homogeneous freezing processes, but it does change the amount of heterogeneous IN, and

reduces the non-physical increase in cloud ice observed in Fig. 5 at high latitudes.

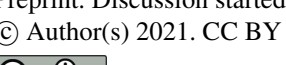



**Figure 5.** Baseline concentration of cloud ice number concentration (a, number of particles per $cm^{-3}$) and fractional occurrence (b) in the period 2010-2030. In the panels c-e) and f-h), the yearly evolution of the differences between GLENS and Baseline (in the 2010-2030 period, as for the panels above) are shown for the three latitudinal boxes separately. In the small panels below, the effect of the changes produced by the aerosols are isolated by plotting the difference between GLENS and the simulations with surface cooling and stratospheric heating but no aerosols.



**Figure 6.** Evolution of the occurrence of homogeneous freezing in clouds (a), the concentration of what the model considers to be IN for homogeneous freezing (Aitken mode sulfate) (b) and the concentration of IN for heterogeneous freezing (coarse mode dust) (c) in GLENS in the first 21 years of simulation, for the three latitudinal band already considered in Fig. 5. On the right, the column mean between 50 and 400 hPa is considered (thick lines), together with the case described in the text with a sensitivity test for Eq. 2.2 (dotted lines).In panel f) the ice number concentration shown in Fig. 5 is compared with that for the sensitivity experiment.





## 5 Radiative effects

Here we investigate whether the changes shown in the previous section produce some signal in the modeled radiative fluxes at the top of the atmosphere, as that would be important in order to determine how significant the effect is. To do so we use the method suggested by Ghan (2013) to separate both the direct radiative effect of the aerosols and the effect of cloud changes

produced by the aerosols. This is possible here as the radiative output of CESM is comprised of both the full radiative fluxes ($\mathbf{F}_{ALL}$) at the top of the atmosphere (TOA), that consider all effects from aerosols, clouds and atmospheric composition, but also of the diagnosed Clear Sky radiative fluxes (where the radiative transfer model calculates the radiative fluxes as if no clouds were present, $\mathbf{F}_{CLEAR}$), Clean Sky radiative fluxes (where the radiative fluxes are calculated as if no aerosols were present, $\mathbf{F}_{CLEAN}$) and their combination ($\mathbf{F}_{CLEAN,CLEAR}$). This allows a separation of the contributions of the added aerosols and

their effects for both the longwave (LW) and shortwave (SW) components: the direct forcing produced by the aerosols direct interaction with the radiation ($\Delta\mathbf{F}_D$) can be defined as

$$\Delta\boldsymbol{F}_D = (\boldsymbol{F}_{ALL}^{GLENS} - \boldsymbol{F}_{CLEAN}^{GLENS}) - (\boldsymbol{F}_{ALL}^{BASELINE} - \boldsymbol{F}_{CLEAN}^{BASELINE}) \tag{5}$$

and the forcing produced by the changes in clouds ($\Delta\mathbf{C}$) can be defined as

$$\Delta\boldsymbol{F} = (\boldsymbol{F}_{CLEAN}^{GLENS} - \boldsymbol{F}_{CLEAN,CLEAR}^{GLENS}) - (\boldsymbol{F}_{CLEAN}^{BASELINE} - \boldsymbol{F}_{CLEAN,CLEAR}^{BASELINE}) \tag{6}$$

As Ghan (2013) notes, normally these fluxes are just an estimation of the real components of the forcing, as to properly estimate them it would be necessary to maintain surface and tropospheric temperatures fixed between the two cases to avoid changes in the radiative emissions of the troposphere, which would be at different temperature in the two cases. In our case, however, the GLENS simulations have been performed on purpose to maintain similar global surface temperatures as the Baseline 2010-2030 period: some small regional temperature differences are still present (Tilmes et al. (2018a)), but overall this can be

considered a minor factor and the estimated forcing quite robust if we are comparing the radiative fluxes always against the 2010-2030 period. In the case of cloud changes, this further assures that we are not counting effects produced by the surface warming, which might also locally modify cloud cover.

In Fig. 7 we show the global evolution of $\Delta\mathbf{F}_D$ and $\Delta\mathbf{C}$ in GLENS. The aerosol direct radiative effect is linear, positive in the shortwave (implying a cooling) as they reflect incoming sunlight and negative in the longwave (implying a warming) as

they absorb and re-emit longwave radiation, with an overall negative effect, as expected. We show the latitudinal breakdown of the fluxes in panels 7b-c normalized by the stratospheric AOD produced by the $SO_2$ injections for the initial period (we pick 2026-2035 to avoid the very first few years when the algorithm that determines the $SO_2$ injections is still converging, see Kravitz et al. (2017)) and the last twenty years (2081-2100). A partial non-linearity can be observed between the two periods. This can be explained by the slight increase in stratospheric sulfate aerosol during the whole period of analyses (see

Visioni et al. (2020a), the effective radius in the stratosphere in GLENS grows from 0.4 $\mu$m to over 0.5 $\mu$m as a result of the increasing injection rates resulting in more coagulation of $SO_2$ with pre-existing particles). Larger aerosols scatter slightly less efficiently, but they absorb more LW radiation. The changes in upper tropospheric aerosols described before do not influence these radiative fluxes, as they are, in proportion, negligible compared to the stratospheric increase. Looking at the $\Delta\mathbf{C}$ fluxes,





however, we can see that the behavior of the LW $\Delta C$ shows a very different behavior compared with the SW. This effect is
positive in the first $\simeq 30$ years, and then negative afterwards, whereas the SW forcing is always positive. The SW forcing is
easily explainable as directly connected to the presence of the aerosols above: if less sunlight reaches the troposphere, because
it is partially reflected in the stratosphere, the same clouds would appear as to be less reflective, hence the positive sign of the
SW (cloud masking). This doesn't apply to the LW, as globally the surface temperatures are the same, but can be explained
by noting that the main contributor to LW trapping by clouds is upper tropospheric ice (Fusina et al. (2007)). Since, in the
first decades, mid-latitudinal ice clouds increase as shown in Fig. 5 because of "more" dust aerosols acting as IN, they would
trap more outgoing LW radiation. When tropical ice clouds start decreasing because of the dynamical mechanism produced
by stratospheric heating, the positive bump is erased by the negative forcing (less ice clouds, more LW radiation escaping to
space) already analyzed by Visioni et al. (2018a). This is further confirmed by the latitudinal breakdown of $\Delta C$ in panels 7e-f,
where the (normalized by stratospheric AOD) forcing in the LW goes from positive, in the first 10 years, to negative in the last
385  20.

## 6  Conclusions

In this work we have identified the presence of some weaknesses in the 3-mode modal approach (MAM3) used in CESM1(WACCM)
when a large presence of aerosols settles down from the stratosphere, as under stratospheric sulfate aerosol injection, which
results in some artificial changes to cirrus clouds. MAM3 only separates the species in three modes depending on their size by
treating all aerosol species as the same. When a large amount of sulfate is produced (mostly in the coarse, largest mode) in the
stratosphere following the injection of $SO_2$ in large quantities, the particles slowly descend upon the troposphere, where they
are quickly removed. However, the size of the Coarse mode particles is different (smaller) compared to that of the Coarse mode
particles already present in the troposphere, whose source is at the surface: therefore, MAM3 "sees" an abrupt decrease in all
aerosol size in each gridbox in the upper troposphere for the coarse mode, resulting in smaller settling velocities for aerosol
species that would not be otherwise affected in the real world. This effect results in an increase in the mass of particles of all
species at those altitudes even if there would be no natural causes for it to happen. The effect is small, and its direct effect on
the cooling produced by the stratospheric aerosols negligible. However, the unnatural addition of dust in the upper troposphere
results in more particles that the model can use as solid ice nuclei for the freezing of ice particles in clouds. This effect is
particularly evident at mid-and-high latitudes, where the low relative humidity and lack of aerosols make other ice formation
processes more scarce in normal conditions. The formation of these ice crystals, as simulated by the model, does produce a
noticeable change in the cloud forcing, especially in the first years, when the effect from the incorrect presence of the dust
aerosols is large compared to other, dynamical effects that tend to reduce the amount of ice crystals in the upper troposphere
when a warming of the stratosphere is present.

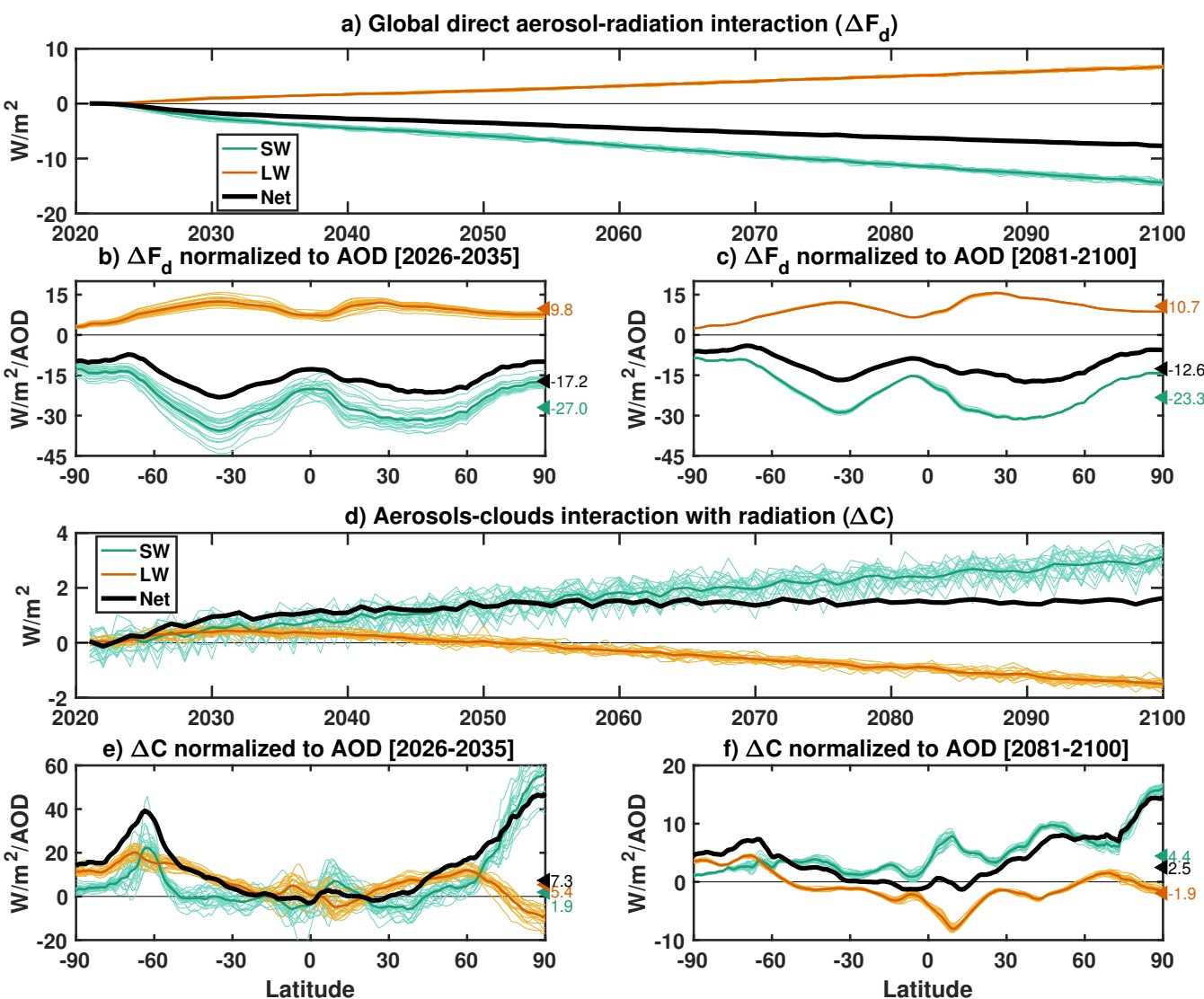

**Figure 7.** Global mean evolution of TOA radiative fluxes in the LW (orange), SW (green) and Net (black) for the forcing directly produced by the aerosols $\Delta \mathbf{F}_D$ (a) and by the clouds as changed by the presence of the aerosols ($\Delta \mathbf{C}$) (d). In panels b-c and e-f, respectively, the latitudinal breakdown of the global fluxes in two periods (2026-2035 and 2081-2100) is shown for the various components. On the right of each panel, the global mean in the considered period is show with a triangle of the same color, and the value on the right. Lighter lines show the single ensemble realizations, while the thicker lines show the ensemble mean.





Given the set-up of the GLENS experiments, this effect is counteracted by the presence of the algorithm which determines how much $SO_2$ is needed every year to counteract the effect of the increased emissions in RCP8.5: even if the radiative imbalance were to be large, the algorithm would just prescribe more $SO_2$ to be injected, therefore resulting, overall, in the same global mean temperatures as if the error in the ice cloud formation was not there. In the sensitivity simulations produced for

Section 4, which reduced the amount of ice clouds incorrectly formed by the model, the cumulative amount of $SO_2$ injected in the first 20 years was 131 Tg-$SO_2$ (6.2 Tg-$SO_2$/year), whereas the same period in the default simulation had a cumulative amount of $SO_2$ of 154 Tg-$SO_2$ (7.3 Tg-$SO_2$/year): considering that, aside from the change in Eq. 2.2, the two simulations were otherwise the same, we can assume that this difference arises from the changes in ice clouds as simulated by the model. Overall then, if we assume this effect is constant throughout the whole simulation period, it would account for an cumulative

injected 88 Tg-$SO_2$, in the entire century: while this amount might seem large, it accounts for less than two years of anthropogenic emissions of $SO_2$ at present (Visioni et al. (2020b)). This, furthermore, assumes a very large injection to overcome a considerable amount of warming (over 4 °) over the entire century. More moderate mitigation scenarios would require far less cumulative injections of $SO_2$ (for instance, see Tilmes et al. (2020)).

This doesn't imply that the effect can be ignored, but suggests that going forward when using a modal approach to aerosol microphysics, simulations where large amount of sulfate is present in the stratosphere should treat sulfate aerosols in a separate coarse mode that is not internally mixed with the other species compared (dust, sea salt, black carbon, etc.). There could be various applications where this observation might be useful: SAI is one example, but the simulation of explosive volcanic eruptions is another case where it could be useful: for instance, Schmidt et al. (2018) used the same CESM1(WACCM) model described

here to estimate the global volcanic radiative forcing in the last 45 years, and ran into a similar observation as found in this paper, but did not find an explanation for it. The mechanism that produced, in their simulations, an increase in ice particles in the upper troposphere is definitely the same we have encountered here, and would explain some of the LW forcing changes diagnosed in their simulations with a similar method (see Fig. 4 and 5, Schmidt et al. (2018)). More generally, it would be crucial to properly represent the upper troposphere in the case of volcanic eruptions to verify the influence of volcanic eruptions on ice

clouds as observed by some studies in the observational record (see for instance Friberg et al. (2015)). Future modeling efforts aimed at better understanding the climatic effect of volcanic eruptions, such as the Volcanic Model Intercomparison Project (Clyne et al. (2021)) or the Interactive Stratospheric Aerosol Model Intercomparison Project (Timmreck et al. (2018)) should take this into account and consider how that might change some of their results, since models with similar modal approaches are present in both.


Lastly, this observation would also be crucial in studies that aim to combine sulfate injections with the artificial seeding of upper tropospheric ice clouds with solid nuclei, to increase the size of the ice crystal and make them sediment faster (cloud seeding, Gasparini et al. (2020)). Cao et al. (2017), for instance, proposed a combination of the two methods to stabilize global temperatures and precipitation: such simulations, performed with CESM1(WACCM) or any model with a similar microphysical

approach would not give meaningful results. Our study shows that more care should be given to make sure that the climate



models used for simulating sulfate geoengineering application are not applied outside of the parameters for which the models will give reliable answers. Most of the time, models are also compared against volcanic events to make sure they are properly simulating the results, but not all effects are immediately visible in those cases as compared to a long term geoengineering simulation, and a comparison of cloud changes after a volcanic eruption may be complicated by various factors (Friberg et al. (2015)).


*Data availability.* All simulations analyzed in this work are available online. All GLENS simulations data are available at https://doi.org/10.5065/D6JH3JXX. Data from the stratospheric heating simulations used for Fig. 5 are available at https://ecommons.cornell.edu/handle/1813/76871.

*Author contributions.* DV performed the analysis and wrote the manuscript. ST, CB and MJM contributed to the analysis and with the interpretation of MAM3 and ice formation processes results. DGM, BK and YR offered helpful suggestions on the manuscript and analysis.


*Competing interests.* The authors declare no competing interests

*Acknowledgements.* Support for DGM was provided by the National Science Foundation through agreement CBET-1818759. Support for DV was provided by the Atkinson Center for a Sustainable Future at Cornell University. Support for BK was provided in part by the National Sciences Foundation through agreement CBET-1931641, the Indiana University Environmental Resilience Institute, and the Prepared for Environmental Change Grand Challenge initiative. The Pacific Northwest National Laboratory is operated for the U.S. Department of Energy by Battelle Memorial Institute under contract DE-AC05-76RL01830. The CESM project is supported primarily by the National Science Foundation.




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
