# Peer review of "Limitations of assuming internal mixing between different aerosol species: a case study with sulfate geoengineering simulations"

_Atmospheric Chemistry and Physics, 2021_

## Author Comment (AC2)

**Reviewer's comments are in bold.** Authors' responses are in blue.

**Review of "Potential limitations of using a modal aerosol approach for sulfate geoengineering applications in climate models" by Daniele Visioni et al. submitted to ACP, https://doi.org/10.5194/acp-2021-678**

**Overview: This study examines how well a modal aerosol model represents tropospheric aerosol loadings in a stratospheric sulfate geoengineering model. The authors demonstrate that the internal mixing assumption of the modal model degrades the simulation of upper tropospheric ice clouds. The overall conclusion is that care needs to be taken when simulating the evolution of stratospheric aerosols with a modal model where the resultant aerosols probably are not internally mixed in actuality. Cases noted are stratospheric geoengineering as in the GLENS runs, and possibly when simulating large volcanic eruptions. The authors also note that this could be an issue in modeling a geoengineering scenario consisting of cirrus cloud thinning in conjunction with increasing stratospheric aerosols via SO2 injection. Overall, it's an interesting paper requiring minimal revision, but requires very careful reading to understand all the points that are important. Figure 1 needs to be revised (see comments below) and I suggest the authors make sure the figure captions fully explain what is shown in the multi panel figures.**

We thank the reviewer for their supportive comments and insightful suggestions for improvements. We address the specific comments below.

**Specific comments:**
**1) (Minor) Crutzen (2006) is not the first study to suggest stratospheric aerosol enhancement to counter global warming. NAS (1992), Panel on Policy Implications of Greenhouse Warming, Policy Implications of Greenhouse Warming: Mitigation, Adaptation, and the Science Base. Washington, DC: Natl. Acad. Press could be referenced, or even one could go back to the original work by Budyko (Budyko, M. I. (1969). The effect of solar radiation variations on the climate of the Earth. Tellus, 21(5), 611–619. https://doi.org/10.1111/j.2153â   3490.1969.tb00466.x)**

Thank you for the suggestion, we have added the reference to the study from Budyko.

**2) Line 105-114: The short description of the GLENS runs is not entirely clear, and there seems to be differences between the project documentation page (https://www.cesm.ucar.edu/projects/community-projects/GLENS/) and what is detailed here. The project page says 20 member ensembles, while this says 21. Make it clear that the 2010-2030 runs are the control runs with no geoengineering. Also, the project page says that 3 members without SAI go to 2100, while this text says 4. It may be that the description here is correct, and the project web page is wrong, but the project web page agrees with the Tilmes et al., 2018a reference.**

Thank you, we have tried to make the description clearer. The reviewer is right that the documentation page only mentions 20 ensemble members, but after Tilmes et al. 2018 one further ensemble member was ran, due to the need to include some more variables unavailable on the original runs for a project unrelated to this paper, and we also included that run here. We have tried to specify this now, and will work to update the documentation page too.

**3) Figure 1:  The caption refers to a black box indicating an area, and then green boxes are mentioned.  I can't tell what is meant there, because I don't see a black box, but rather a black hat shape almost (at 110 hPa, lines from 90S-60S, and 90N-60N, and then at 60 hPa from 30S to 30N). And, there are 3 green boxes.  So, perhaps instead you can say in the caption what the area of consideration is,  does black box mean everything under the black line, or everything above it?  And, the letters (a,b,c,d) are on the figures, but no described in the caption (just delete them).  The bottom left and center figures have text that overlaps.  The caption should also say what the blue and red lines are.  (they are labeled in the figure as baseline and GLENS for the middle panels, the caption should say that too, and also if that is valid for the rightmost panel).  Defining the Baseline case in the caption would be useful as well.  Also, define in the caption what is meant by "last 20 years" for the rightmost panels.  Is that 2100-2080 in the simulation?  And is red RCP8.5 (no SAI) and Blue RCP8.5+SAI?  And what mode it is should be labeled on each panel (rather than calling things SO4a etc). The figure labeling is a little confusing.**

We thank the reviewer for their suggestions, and we apologize for the confusion! We have tried to improve the figure now and made the caption clearer. The new caption reads:

"*Mass concentration (in μg per kg of air) of Coarse mode species (Sulfate (SO4), Dust (D) and Sea Salt (SS), from top to bottom). On the left, panels a) indicate the Baseline (2020-2029) zonal mean concentration for the respective species, with the black and green boxes indicating the area considered in the averages for the central panels (panels b and c). The uppermost limit of the green boxes and the lowermost limit of the black boxes coincide. Panels b) and c) for each species show the annual evolution of the concentration in the black (panels b) and green (panels c) boxes, with lighter lines showing the single ensemble realizations, and the thicker lines show the ensemble mean; red lines indicate the Baseline simulations, while blue lines indicate the GLENS simulations. On the right, panels c) show the vertical profiles in both cases (Baseline, in red, and GLENS, in blue) for the period 2080-2099. Black dashed lines in the c) and d) panels indicate the periods of analyses for the a) and d) panels, respectively; the same lines in the d) panels indicate the altitude of analyses in the b) and c) panels.*"

**4) Line 218/219 says "However, the vertical profiles of tropical concentration for the three species seem to exclude that."  This isn't shown in figure 1, should this include a reference to figure 3?**

It does show that in both Figure 1 (panels d) and Figure 3. We have updated the phrase to better reflect that.

**5) Figure 4:  what do the dashed lines in the panels mean?  I've assumed in the top 2 panels that tells what is being plotted in the panels below.  But I've no clue what the line at ~400 hPa means.**

We assume the reviewer is talking about Fig. 5 as there are no dashed lines in Fig. 4 at 400 hPa. We have removed the lines since they belonged to another piece of analyses we then didn't use. Thanks for pointing it out.

---

## Author Comment (AC3)

Review on "Potential limitations of using a modal aerosol approach for sulfate geoengineering applications in climate models" by Visioni et al.

This study investigates the potential issues of using a modal aerosol scheme for simulating geoengineering sulfate aerosols. Simulation results from the CESM1 Geoengineering Large Ensemble (GLENS) project are analyzed and impacts on aerosol concentrations and radiative fluxes are quantified. This is a useful and important contribution which points out the importance of carefully designing aerosol configuration (e.g., mixing state, mode standard deviation, size range), particularly for representing the unconventional cases of geoengineering stratospheric aerosols.

My main comment is that the issues are not due to the modal aerosol scheme itself (as pointed out in the title of this paper), but to the negligence in the scheme design. Modal schemes assume internal mixing between aerosol species within a single aerosol mode, and this is the case of bin schemes which assume internal mixing between aerosol species within a single aerosol bin. Different aerosol modes should be created for tropospheric coarse dust/sea salt and for stratospheric coarse sulfate as they have vastly different properties. We cannot simply lump these in the same aerosol mode. We note that the aerosol schemes used in global climate models often have to be simplified (minimalized) due to the consideration of computational efficiency. For the comparison, the bin scheme (CARMA in CESM) uses a group of bins for pure sulfate and another group of bins for mixed aerosols (POM, BC, dust, sea salt, sulfate internally mixed in a bin). The modal scheme (MAM4 in CESM) has done similarly to design the primary carbon mode specifically for carbonaceous aerosols (POM/BC). Currently efforts are underway (to develop MAM5) by adding a new mode for stratospheric sulfate separated from the tropospheric coarse mode for dust/sea salt.

Based on the above comments, major revision of the manuscript is required before the paper can be accepted by ACP, including the title, abstract, and conclusions in the text to make it clear that the issues are not due to the modal scheme itself and call for more careful design of mode (bin) structures.

We thank the reviewer for their supportive comments, and for making a very important point regarding our manuscript. We have modified all necessary parts as the reviewer suggested.

The title has been changed to: "*Limitations of assuming internal mixing between different aerosol species: a case study with sulfate geoengineering simulations*"

The abstract has been changed to (changes in bold, line and page referring to the revised version)

(lines 1-5, p1) "Simulating the complex aerosol microphysical processes in a comprehensive Earth System Model can be very computationally intensive and therefore many models utilize a modal approach, where aerosol size distributions are represented by observations-derived lognormal functions, **and internal mixing between different aerosol species is often assumed**" (line 6, p1) "In this work we show specific conditions under which the current approximations used in **some** modal approaches might yield some incorrect answers"

In the Conclusions, some remarks have been added:

(line 392, p19) "MAM3 only separates the species in three modes depending on their size, by treating all aerosol species as the same **(internal mixing assumption)**."

(line 423-424, p. 21) "A similar approach has already been used to include an additional primary carbon mode in MAM4 (Liu et al., 2016) in order to account for processes that affect the microphysical properties of primary carbonaceous aerosols in the atmosphere."

**Minor comments:**

Line 50-54, please refer to Riemer et al. for more accurate definition of aerosol mixing state: Riemer, N., Ault, A. P., West, M., Craig, R. L., & Curtis, J. H. (2019), Aerosol mixing state: Measurements, modeling, and impacts. *Reviews of Geophysics*, 57, 187–249. https://doi.org/10.1029/2018RG000615.

**Added.**

Line 101. "observed" change. I am sure the "observed" means the "identified", "noticed" or "found" in your study, not change from observations (in the field campaign or laboratory). To avoid confusion, please use another word. This same is true for "observed" and "observation" in many places in the following text.

We have modified "observed" in "identified" as suggested by the reviewer. We have identified all cases in the manuscript where the term "observed" or "observation" was used improperly, and modified them all accordingly.

Line 131-137. Some statements are not clear: "only one type of particles for each mode is considered"; "Therefore, each particle species shares the same radius and number concentration per each mode,". Because of internal mixing of aerosol species in an aerosol mode, composition of aerosols within one mode is the same, as well as size. Refer to the above Riemer et al. paper for the definition of aerosol mixing state.

We have simplified the phrase in "Compared to the more computationally-demanding MAM version with 7 aerosol modes (MAM7), in MAM3 all aerosol species are considered internally **mixed within each of the three modes, thus sharing composition and size distribution**. The mass for the single species has to be conserved, both globally and locally, and thus can only change in each gridbox if particles are moved from one gridbox to the other, either because of air mass movement, or because of gravitational settling or other tropospheric removal processes."

Line 187. "which can then oxidize and form sulfate particles of sulfuric acid in the smaller (Aitken) mode." The sentence is awkward. Might change to "which can be oxidized to form sulfuric acid and then sulfate particles by condensation in the smaller (Aitken) mode."

We thank the reviewer for the suggestion. The phrase has been changed accordingly.

Line 234. "In MAM3, all aerosol species are assumed to be internally mixed." should be "In MAM3, all aerosol species *within an aerosol mode* are assumed to be internally mixed."

**Added, thank you!**

Line 252-253 and Figure 1. The increases of dust and sea salt in UTLS regions in GLENS is likely

due to the renaming (transfer) of the accumulation mode dust and sea salt (along with stratospheric sulfate) to the coarse mode. Because of the small standard deviation of coarse mode, dust and sea salt are accumulated and increased there.

If the observed change was due to a transfer from one mode to the other, however, there would have to be a similar reduction in the accumulation mode in those species. Figure 3, on the other hand, shows that accumulation mode dust and sea salt are slightly higher above 300 hPa in GLENS compared to the Baseline case, and little change is observable below that. This lead us to exclude a significant contribution from a renaming of accumulation mode particles into coarse mode.

Line 282. "known as immersion or heterogeneous freezing". Use "heterogeneous nucleation". Immersion is just one of the mechanisms of heterogeneous nucleation of ice.

Changed, thank you for the suggestion.

Line 364. "positive in the shortwave (implying a cooling)". In Figure 7, it is shown to be negative.

Sorry, that should have said "negative in the shortwave". We have corrected it.

---

## Author Response (AR2)

**Response to Reviewer #1**

**Reviewer's comments are in bold.** Authors' responses are in blue.

**This is a reasonable paper suitable for ACP. The authors have addressed most of the comments but improvements and corrections are needed in a couple of places (see below).**

Thank you! We have addressed the single points below.

**1. Reviewer #2, Specific comments #1. The reviewer recommended the citation of NAS (1992) and Budyko (1969). However, the authors cited a later paper by Budyko (1978) without explanation. Based on the titles, I think that NAS (1992) and Budyko (1969) are more suitable references here.**

Thank you for pointing this out. We have substituted the Budyko (1978) reference with the correct Budyko (1969) one in the manuscript.

**2. Reviewer #2, Specific comments #3 – Fig 1 caption. The revised caption is still not clear enough and there exist errors. The authors should carefully proofread the writings.**
**(1) Caption mentions panels a), b), c), and d) but in the plots, there are multiple panels with "a)" although these "a)" are distinguished by SO4, D, SS. Should try to be consistent throughout the plot, caption, and text.**
**(2) 4th line from bottom: On the right, panels d)?**
**(3) 3rd line from bottom: "c) and d)" should be "c)"?**

We thank the reviewer for their feedback. In the text, we have been more consistent in naming the single panels as in the figure. In the figure, we included two separation lines between the three species for clarity. In the caption we fixed the two mistakes found by the reviewer and carefully checked for consistency throughout.

**Other comments:**
**Line 373: "negative in the longwave" should be "positive in the longwave"?**

Corrected, thank you.

**Response to Reviewer #2**

**Reviewer's comments are in bold.** Authors' responses are in blue.

**The authors have addressed my comments sufficiently. Two minor changes:**
**1. in the abstract (Line 3), change "and internal mixing between different aerosol species is often assumed." to "and internal mixing between different aerosol species within an aerosol mode is often assumed."**

Done!

**2. in the abstract (Line 6), there are two "some" in this sentence. You can remove the second one.**

Done! And thank you for help us improving our manuscript.